# Monitoring Moroccan Honeys: Physicochemical Properties and Contamination Pattern

**DOI:** 10.3390/foods12050969

**Published:** 2023-02-24

**Authors:** Abir Massous, Tarik Ouchbani, Vincenzo Lo Turco, Federica Litrenta, Vincenzo Nava, Ambrogina Albergamo, Angela Giorgia Potortì, Giuseppa Di Bella

**Affiliations:** 1Institut Agronomique et Vétérinaire Hassan II, Rabat 10101, Morocco; 2Department of Biomedical, Dental, Morphological and Functional Images Sciences (BIOMORF), University of Messina, Viale Annunziata, 98122 Messina, Italy

**Keywords:** Moroccan honey, beekeeping, physicochemical traits, organic contaminants, plasticizers, bisphenols, potentially toxic elements

## Abstract

The physicochemical traits and an array of organic and inorganic contaminants were monitored in monofloral honeys (i.e., jujube [*Ziziphus lotus*], sweet orange [*Citrus sinensis*], PGI Euphorbia [*Euphorbia resinifera*] and *Globularia alyphum*) from the Moroccan Béni Mellal-Khénifra region (i.e., Khénifra, Beni Méllal, Azlal and Fquih Ben Salah provinces). Moroccan honeys were in line with the physicochemical standards set by the European Union. However, a critical contamination pattern has been outlined. In fact, jujube, sweet orange, and PGI Euphorbia honeys contained pesticides, such as acephate, dimethoate, diazinon, alachlor, carbofuran and fenthion sulfoxide, higher than the relative EU Maximum Residue Levels. The banned 2,3′,4,4′,5-pentachlorobiphenyl (PCB118) and 2,2′,3,4,4′,5,5′-heptachlorobiphenyl (PCB180) were detected in all samples and quantified in jujube, sweet orange and PGI Euphorbia honeys; while polycyclic aromatic hydrocarbons (PAHs), such as chrysene and fluorene, stood out for their higher contents in jujube and sweet orange honeys. Considering plasticizers, all honeys showed an excessive amount of dibutyl phthalate (DBP), when (improperly) considering the relative EU Specific Migration Limit. Furthermore, sweet orange, PGI Euphorbia and *G. alypum* honeys were characterized by Pb exceeding the EU Maximum Level. Overall, data from this study may encourage Moroccan governmental bodies to strengthen their monitoring activity in beekeeping and to find suitable solutions for implementing more sustainable agricultural practices.

## 1. Introduction

Honey is a sweet product of the bee *Apis mellifera* endowed with very specific physicochemical properties, which make it unique from other viscous solutions, as well as precious healthy and therapeutic properties, mainly related to the presence of enzymes, vitamins, phenolics and minerals, which make it a functional food [1]. However, despite its biological activities, honey is not free of contaminants, and the monitoring of its chemical safety is still crucial today not only for assuring product quality and consumer health protection, but also for preserving the environment with its landscapes and biodiversity.

Bees are exposed to numerous pollutants during their foraging activities: their hairy bodies can easily retain a variety of environmental contaminants, and they can be contaminated through food resources when collecting pollen and nectar from flowers or water [2,3,4]. As a result, xenobiotic residues are transferred into and accumulate in bee products, including honey [5,6], which can be regarded not only as food products but also as reliable indicators of environmental contamination [7,8].

Among organic pollutants, organochlorine pesticides (OCPs) and polychlorinated biphenyls (PCBs) represent substances with a high environmental persistence and detrimental effects on human health, whose production is severely banned or restricted all over the world according to the International Stockholm Convention [9]. Nevertheless, their residues are still found in the environment, in food—honey included—and in biological matrices [10,11,12,13]. While not listed under the Stockholm Convention, polycyclic aromatic hydrocarbons (PAHs) were, in June 1998, one of 16 groups of chemicals listed under the United Nations Economic Commission for Europe (UN-ECE) Convention on Long Range Transboundary Air Pollution (LRTAP), that was signed by 33 countries and the European Commission.

The surrounding environment, with its urban, industrial, and agricultural activities, and even beekeeping [14], is not the only source of contamination. Plastic additives, such as phthalates non-phthalate plasticizers (PAEs and NPPs) and bisphenols (BPs) may also occur in honey, and their presence may be explained not only by the chemical load of the same raw material, due to the ubiquitous presence of such contaminants, but also by the processing involving the direct contact of honey with plastic materials [15,16,17].

To protect the authenticity of honey from adulteration events, precise standards on its physicochemical quality have been provided by the Codex Alimentarius [18], and then, further adjusted by the EU Community [19]. However, the international legislative framework on the chemical safety of such bee products is still fragmentary. Indeed, the International Codex Standard for honey refers only to contaminants such as pesticides and heavy metals and suggests that the respective maximum residue limits (MRLs) and maximum levels (MLs) should comply with the standards set by the Codex Alimentarius Commission for contaminants in food [20]. In this scenario, the EU took the lead in regulating apicultural products in September 2018 by approving the “Technical guidelines for determining the magnitude of pesticide residues in honey and setting Maximum Residue Levels in honey” [21]. On the other hand, for heavy metals that may exhibit potential toxicity to consumers, and other organic contaminants, such as PAHs, the MLs set by Regulation (EC) No. 1881/2006 for a variety of foodstuffs may be (improperly) considered [22].

Similar issues are also experienced when considering process contaminants. Specific migration limits (SMLs) have been applied by the EU to those contact materials containing plastic additives with the potential of leaching into food. SMLs have been set up by the Commission Regulations No. 10/2011 for some PAEs and No. 213/2018 and No. 1907/2006 for bisphenol A and S [23,24,25], and, in the absence of legal limits of process contaminants in food, they are often used, albeit improperly, to obtain an idea of the degree of food contamination.

Since the EU market is a major honey importer and greatly affects the agricultural practices of exporter countries, beekeepers all over the world tend to adopt the mentioned EU legislations as a reference for quality control of honey.

Due to its important floristic, faunal and landscape diversity, Morocco is endowed with an important and unique beekeeping potential, resulting one of the most valuable territories for honey production in the Mediterranean area [26]. Here, beekeeping is a well-rooted tradition and one of the most profitable businesses, thanks to the conspicuous production not only of honey, but also of pollen, propolis, beeswax, and royal jelly. A market overview showed that the Moroccan honey production increased from 4.7 tons to almost 8 tons between 2010 and 2020, with a turnover of around 101 million [27]. Moreover, a vibrant modernization of the sector has started in the last decade, thanks to the leverage effect of the Green Morocco Plan (GMP), the National Initiative for Human Development (NIHD) and, not least, the Moroccan Ministry of Agriculture, setting a referential catalog for high-quality terroir products, including honey, with the final aim to label them under Geographical Indications, Designations of Origin or Agricultural Labels, thus, promoting their consumption [28]. The Béni Mellal-Khénifra region, placed between the High and Middle Atlas mountain ranges and the Tadla Plain, is well known for its rich and varied botanical diversity allowing for significant honey production [29]. Particularly, the region boasts the monofloral Euphorbia honey of Tadla-Azilal, labeled with Protected Geographical Origin (PGI) and produced from *Euphorbia resinifera*, an endemic Moroccan species mainly distributed in Azilal and Béni Mellal areas [30].

A literature overview suggests that, over the past decade, major efforts have been devoted to the characterization of physicochemical, melissopalynological, antioxidant and microbiological traits of Moroccan honey [28,31,32,33,34,35,36,37,38,39,40,41], while very few and fragmentary data are available on contaminants [42,43]. However, the chemical safety of such a product showing a considerable ability to accumulate xenobiotics from the surrounding environment should not be overlooked, also in view of the patchy regulatory framework, as well as the ongoing modernization facing Moroccan beekeeping and its labelled products.

Within this background, the aim of the study was to explore the physicochemical traits and the contamination pattern of monofloral honeys from diverse areas of the Béni Mellal-Khénifra region, including the PGI Euphorbia honey of Tadla-Azilal. Data from the main quality indicators (i.e., moisture, sugars, pH, electrical conductivity, and acidity), as well as from an array of organic chemicals -including regulated (i.e., pesticides, PAEs, NPPs and BPs) and banned/restricted substances (i.e., OCPs, PCBs and PAHs)- and inorganic pollutants (i.e., potentially toxic elements) were employed to statistically evaluate the relation between the quality and safety of honey and the actual production scenarios of the Moroccan region. In addition, the dietary exposure to contaminants derived from the consumption of such honeys was evaluated.

## 2. Materials and Methods

### 2.1. Study Area

The Béni Mellal-Khénifra region is a new Moroccan region created according to the new administrative subdivision of 2015; it represents 4% of Morocco’s national territory and covers 28 088 km^2^, of which more than 65% is mountainous. The region is located in central Morocco, between the High and Middle Atlas and the Tadla Plain, and comprises five provinces: Azilal, Béni Mellal, Fquih Ben Salah, Khénifra and Khouribga (Figure 1).

Thanks to diversified climates and landscapes, this region has a rich natural heritage and high biodiversity, with a significant potential for agricultural development. Indeed, the Béni Mellal-Khénifra region has an urbanization rate (49%) lower than the national average (60.36%), and more than half of the population lives in rural areas (51%), being strongly committed to the agricultural sector, which not only constitutes the major economic activity of the region, but it is also in the process of modernization as required by the GMP [44,45]. However, intense agricultural practices have been seen as a relevant factor for environmental pollution. In this respect, recent studies have reported in diverse areas cases of agricultural soils and groundwater/wastewater intended for irrigation being contaminated, especially in terms of heavy metals and pesticides [46,47,48,49].

### 2.2. Honey Samples

The study was conducted on 12 honey samples produced in 2021 by beekeepers located in diverse provinces from the Béni Mellal-Khénifra region of Morocco. They included *n* = 3 honey samples from *Ziziphus lotus* (i.e., jujube honey) produced in Khénifra, *n* = 3 honeys from *Citrus sinensis* (i.e., sweet orange honey) collected in the Béni-Mellal province, *n* = 3 PGI honeys from *Euphorbia resinifera* (i.e., Euphorbia honey) produced in the Tadla-Azilal area, and *n* = 3 honeys from *Globularia alypum* obtained from the Fquih Ben Salah province. Honeys from this study derived from pooling a given type of honey from different hives of a given area and they were collected in glass jars of ~125 g and stored at room temperature in a dark place until analysis.

### 2.3. Chemicals and Reagents

Analytical standards of *n* = 108 pesticides, *n* = 18 PCBs, and *n* = 13 PAHs were purchased from Sigma-Aldrich (Chicago, Il, USA), Fluka Analytical (Milan, Italy) and Dr. Ehrenstorfer (Augsburg, Germany). For pesticides, deuterated analogues intended as internal standards (ISs) were carbofuran-d_3_, dimethoate-d_6_, atrazine-d_5_, cyprodinil-d_5_, imazalil-d_5_, malathion-d_6_, methiocarb-d_3_, and trifloxystrobin-d_6_. They were all provided by Toronto Research Chemicals (Toronto, CA, USA). For the analysis of PCBs and PAHs, the deuterated analogues were naphtalene-d8, acenaphtene-d10 and phenanthrene-d10, from Cambridge Isotope Laboratories Inc. (Andover, MA, USA). Analytical standards of *n* = 10 PAEs and *n* = 8 NPPs (certified purity ≥96%) were provided by Supelco (Bellefonte, PA, USA). DBP-d4 and DEHP-d4 were the ISs purchased from Cambridge Isotope Laboratories Inc. Analytical standards of *n* = 9 BPs (certified purity ≥97%) were purchased from Sigma-Aldrich (Steinheim, Germany), while the IS ^13^C_12_-BPA was obtained from Cambridge Isotope Laboratories.

Solvents (i.e., acetonitrile, water, and n-hexane, LiChrosolv and Parasol grade) were purchased from Merck (Darmstadt, Germany). The Q-sep QuEChERS extraction kit (4 g MgSO_4_ and 1 g NaCl) and QuEChERS d-SPE (750 mg MgSO_4_, 250 mg of primary and secondary amines PSA and 125 mg C18) were purchased from Agilent Technologies Italia S.p.A. (Milan, Italy).

Overall, the contact of laboratory equipment and solvents with samples, the sample preparation time, and the solvent volumes were mandatorily minimized to significantly reduce the background contamination caused by solvents and laboratory materials. Glassware and stainless-steel instruments were washed with acetone, rinsed with hexane, dried at 400 °C for at least 4 h, and finally wrapped with aluminum foil until analysis. All solvents were tested before use, and due to the negligible levels of background contamination, they were employed throughout the analytical procedures with no further purification.

For the screening of inorganic elements, HNO_3_ (65% *v*/*v*) and H_2_O_2_ (30% *v*/*v*) were of Suprapur grade (Mallinckrodt Baker, Milan, Italy). Ultrapure water (<5 mg/L TOC) was obtained from a Barnstead Smart2Pure 12 water purification system (Thermo Scientific, Milan, Italy). A standard solution of Re (1000 mg/L in 2% HNO_3_) was provided by Fluka (Milan, Italy) and employed as IS. Single-element standard solutions of inorganic elements such as K, Na, K, Mg, Ca, Fe, Mn, Cr, Co, Ni, Cu, Zn, Al, Pb, Cd and As, at a concentration of 1000 mg/L in 2% HNO_3_ (Fluka, Milan, Italy) were used to prepare multielement stock standard solutions. To avoid undesirable cross-contamination, laboratory glassware and plastic instruments necessary for sample collection, handling, and storage, as well as polytetrafluoroethylene (PTFE) digestion vessels, were washed with 5% HNO_3_ before use.

### 2.4. Physicochemical Parameters

Moisture (%) and total soluble solids (TSS) represented by soluble sugars content and expressed as °Brix, were obtained from the tables of correspondence between a given water content/°Brix and the refractive index calculated for each sample at 20 °C. If the index was not determined at a temperature of 20 °C, the correction temperature was considered, and the result was reduced to a temperature of 20 °C [50].

Free, combined, and total acidity were determined by the titrimetric method proposed by Bogdanov and colleagues [50]. Briefly, the titration of the honey sample (10 g diluted with 75 mL of distilled water) was carried out with 0.05 N NaOH to pH 8.5 (free acidity). Then, a 10 mL volume of NaOH was added and titrated again with 0.05 N HCl to pH 8.3 (combined acidity). Total acidity was calculated obtained by the sum of free and combined acidities.

The pH and electrical conductivity for every honey sample were determined by a pH/conductivity meter. Approximately 10 g honey was dissolved in 75 mL distilled water and the pH and electrical conductivity were measured. For electrical conductivity, the quantity of honey to be weighed was calculated using the following Equation (1):(1)$$M=20\times \frac{100}{100-A}$$*M*: mass of honey (g); *20*: is the theoretical nominal mass of honey; *A*: water content in %.


The ashes were obtained by drying 5 g of every honey sample at 600 °C until constant weight, according to the AOAC protocol [51]. For the determination of minerals (K, Mg, Na, and Ca) and essential trace elements (Mn, Fe, Cu, Zn, Se, Cr, Co and Ni), see Section 2.7.

### 2.5. Pesticide, PCB, and PAH Residues

For extraction of pesticides, PCBs, and PAHs from honey samples, the procedure of Saitta et al. [52], with some modifications, was used as described below. Briefly, 10 g of honey was weighed into a tube with 10 mL of water and 10 mL of acetonitrile, and vortexed for 5 min. Then, Q-sep QuEChERS kit and d-SPE (described in Section 2.2) was added and centrifuged for 5 min (5000 rpm). At the end, 5 mL of the organic phase was collected, reduced to 1 mL in a rotary evaporator at 30 °C and reduced to a volume of 0.5 mL volume under a stream of nitrogen. Before instrumental analysis, a known amount of every IS was added to every sample. The multiresidue screening was performed by a Thermo Scientific Trace GC Ultra coupled with a TSQ Quantum XLS triple quadrupole mass spectrometer equipped with a TrisPlus RSH automatic sampler. Separation conditions, and mass spectrometry (MS) details are available in our previous work [6]. Compound identification occurred by comparison of their retention times and mass spectra with those of corresponding commercial standards. The quantitative procedure was carried out in multiple reaction monitoring (MRM) mode, exploiting the IS normalization. The MRM transitions, as well as the main figures of merit of analytical validation are reported in Appendix A. Every honey sample was monitored in triplicate, along with analytical blanks.

### 2.6. PAEs and NPPs Residues

The extraction of plasticizers from the various honey samples was performed according to a method already reported in Liotta et. al. [6], with some modifications. Briefly, 5 g of honey was weighed into a tube and extracted with 10 mL of acetonitrile. Then Q-sep QuEChERS was added and centrifuged for 5 min (5000 rpm). Approximately 2 mL of the organic phase were collected, evaporated to 1 mL in a rotary evaporator at 30 °C and finally reduced to a volume of 0.5 mL volume under nitrogen stream. Before instrumental analysis, a known amount of DBP-d4 and DEHP-d4 was added to every sample. The multiresidue screening was carried out by a gas chromatography system (GC-2010, Shimadzu, Japan) equipped with an autosampler (HT300A, HTA, Italy) and coupled to a single quadrupole mass spectrometer (QP-2010 Plus, Shimadzu, Japan) according to the operating conditions already described in our previous work [53]. Identification of PAEs and NPPs occurred by comparison of their retention times and mass spectra with those of corresponding commercial standards, while the quantitative assay was performed in SIM mode, considering the base peak ion out of three characteristic mass fragments for each target analyte (Appendix A) and using the IS normalization. The parameters of acquisition, as well as the main figures of merit of analytical validation are reported in Appendix A. Measurements were conducted in triplicate for every sample, alternated with analytical blanks.

### 2.7. BP Residues

For the extraction of the nine bisphenols, the method already proposed by Liotta et al. [6] with some modification, was applied. Briefly, 5 g of honey was placed in centrifuge tubes with 10 mL of ultrapure water and 10 mL of acetonitrile, and vortexed for 5 min. Then, 4 g of MgSO_4_ and 1.5 g of NaCl were added. The obtained mixture was vortexed for 5 min, and centrifuged at 4000 rpm for 10 min. Then, 5 mL of supernatant was added to the QuEChERS d-SPE cleaning tube, vortexed and placed in centrifuge at 4000 rpm for 10 min. Hence, 1 mL of supernatant was recovered and filtered by 0.22 µm nylon filter and analyzed by HPLC-MS/MS. Analysis was performed on an LC apparatus (Prominence UFLC XR system, Shimadzu, Kyoto, Japan) consisting of a controller (CBM-20 A), binary pumps (LC-20AD-XR), degasser (DGU-20A3R), column oven (CTO-20AC), and autosampler (SIL-20 A XR). An electrospray ionization (ESI) source interfaced the LC system to a triple quadrupole mass spectrometer (MS) (LCMS-8040, Shimadzu, Kyoto, Japan). Data were acquired in MRM mode and the resulting ion transitions were used for the identification and quantification (internal standard method) of BPs. MRM transitions and the main figures of merit of analytical validation for every target analyte are reported in Appendix A. Every honey sample was monitored in triplicate along with analytical blanks.

### 2.8. Inorganic Elements

Mineralization of honey samples was carried out following the method proposed by Di Bella and coworkers [54]. About 0.5 g of each honey sample was weighed, and 1 mL of IS at 0.5 mg/L was added. The samples were digested with 7 mL of HNO_3_ (65%, *v*/*v*) and 1 mL of H_2_O_2_ (30%, *v*/*v*) in a microwave ETHOS 1 digestion system (Milestone, Bergamo, Italy) using the following instrumental parameters: 15 min at 1000 W up to 200 °C, 15 min at 1000 W at 200 °C. The digested samples were conveniently diluted with ultrapure water and their analysis was carried out by means of a single quadrupole inductively coupled plasma-mass spectrometer (ICP-MS, iCAP-Q, Thermo Scientific, Waltham, MA, USA) according to the operating conditions already reported in our previous studies [5,55,56]. All samples were processed in triplicate along with analytical blanks. The analytical validation of the ICP-MS method is reported in Appendix A.

### 2.9. Statistical Analysis

Statistical analysis was carried out using the SPSS 13.0 software package for Windows (SPSS Inc., Chicago, IL, USA). Initially the non-parametric Kruskal–Wallis test was applied on log-transformed data to assess differences between honey samples, with a statistical significance at *p* < 0.05. Subsequently, a Principal Component Analysis (PCA) was conducted on a starting data matrix where the cases (12) were the analyzed honey samples and the variables (54) were the values of physicochemical parameters, as well as pesticides, PCBs, PAHs, plasticizers, BPs residues and element concentrations that were higher than their respective LOQs. When concentrations were below the limit of quantification (LOQ), these were replaced with half the limit of detection (LOD/2). Then, the data set was normalized to achieve independence of the different variables scale factors and a PCA was performed to evaluate the differentiation of honey samples in relation to the different production context and/or floral origin according to the investigated variables.

### 2.10. Assessment of the Dietary Exposure to Contaminants

To evaluate the health risks of organic and inorganic contaminants derived from the intake of Moroccan honey, the relative estimated daily intakes (EDI) were calculated by multiplying the mean contaminant concentration found in every sample (mg/Kg or µg/Kg) by the amount of honey consumed daily (g/day) and dividing the obtained result by the consumer’s body weight (Kg_bw_). Hazard quotient (HQ), which is the ratio between a given EDI and the corresponding oral reference dose (RfD) proposed by the U.S. Environmental Protection Agency (US EPA,) was also employed to assess the plausibility of risk. An HQ (dimensionless) >1 entails a high non-carcinogenic risk.

## 3. Results and Discussion

### 3.1. Physicochemical Parameters

#### 3.1.1. Moisture, TSS, Acidity and pH

The values of moisture, TSS, free, combined, and total acidity, and pH of Moroccan honeys are shown in Table 1.

The moisture of honey is strictly related with the harvest time and practices performed by beekeepers, and, not least, the level of honey maturity reached in the hive [57]. This parameter influences honey flavor, color, density, and viscosity, and determines its stability and granulation during storage [58]. In the honeys investigated, moisture values were always below the maximum limit (20%) set by the Codex Alimentarius and EU standards [18,19]. This may be due not only to the correct time of extraction by Moroccan beekeepers, but also to the current use of modern hives with better moisture control. Specifically, moisture values ranged between 14.93–16.57%, thus indicating a good degree of maturity of all products. Despite the small variability, the upper and lower moisture values, represented respectively by the jujube honey from Khénifra and the sweet orange honey produced in Béni Mellal province, were significantly different (*p* < 0.05), which may suggest a variation in such parameters in relation to the climatic conditions [59].

For TSS, minimum/maximum values of 82.67 °Brix/85.83 °Brix were found respectively in sweet orange and *G. alypum* honeys, fall within the acceptability range (78.8 and 85 °Brix). Indeed, TSS values are known to decrease with the increasing concentration of starch, molasses, glucose, and distilled water. As a result, this parameter is inversely related to the moisture content and useful in the detection of adulteration events [60].

Honey acidity and pH are correlated with each other, being dependent on the level of organic acids and enzymatic activity in honey. As a result, their variation in honey samples could be attributed to the floral origin rather than the environmental context [57].

These physicochemical parameters are generally intended as a marker of honey freshness since the higher the acidity and the lower the pH, the better the environment that inhibits microorganism growth [61]. In the present study, all products showed acidity values below the EU standards 50 meq/kg [14], thus suggesting the absence of undesirable fermentation and/or bacterial spoilage. Specifically, free, and total acidity values ranged from 15.41 meq/kg and 16.39 meq/kg in sweet orange honeys (Khénifra) to 39.28 meq/kg and 39.88 meq/kg in *G. alypum* honeys (Fquih Ben Salah), thus yielding significantly different results in these types of honey (*p* < 0.05).

All samples analyzed showed an acidic character. Although non significantly different (*p* > 0.05), pH values were in accordance with the acidity values and varied probably due to the different floral origin. In fact, the lowest pH was observed in *G. alypum* honey (3.98), while the highest values were found in sweet orange and jujube honey (4.24).

Overall, the physicochemical traits discussed in this study were in line with those reported for the recent production of Moroccan honey from Middle Atlas, also with the same floral origin [28,32,33,38,40,41,58].

#### 3.1.2. Electrical Conductivity, Ash, and Mineral Content

The electrical conductivity, ash, and mineral profile (K, Ca, Na, and Mg) and essential trace elements (Mn, Fe, Zn, Cu, Se, Cr, Co, and Ni) of investigated Moroccan honey are reported in Table 2.

The electrical conductivity of the honey is closely related to the concentration of minerals and organic acids and its assessment is useful in the discrimination between blossom and honeydew honeys. In fact, such a parameter tends to be higher in honeydew honeys and it varies in relation to the same honeydew content. The Codex Alimentarius and EU legislation require blossom honeys to have conductivity values not higher than 800 µS/cm [18,19]. As a result, honeydew honeys generally show higher values than 800 µS/cm. Additionally, in monofloral honeys, this parameter shows great variability according to the floral origin [62,63].

In Moroccan honeys, conductivity varied from 157.00 µS/cm to 633.67 µS/cm, respectively in sweet orange and *G. alypum* honey, which consequently gave significantly different results (*p* < 0.05). On the other hand, jujube and *Euphorbia* honeys showed intermediate and non-significantly different conductivities (respectively, 381.33 µS/cm and 362.67 µS/cm, *p* > 0.05). All values were below the maximum limit (800 µS/cm) set by the Codex Alimentarius and EU standards for such parameter [18,19].

Differently from conductivity, there is no specific legislation on maximum level of ash, minerals, and trace elements content in honey, which, consequently, are not yet considered as a quality parameter by either the Codex Alimentarius or the EU. However, they are very important quality markers of honey, reflecting both the floral source of honey as well as its environmental context of production [64].

Ashes followed the same trend of electrical conductivity in investigated honeys, ranging from 0.34 g/Kg in sweet orange honey from Béni Mellal to 1.17 g/Kg in jujube honey from Khénifra (*p* < 0.05). In terms of concentrations, similar considerations could also be made for the element profile.

Considering minerals, K was the most abundant mineral in all honeys analyzed, followed by Ca, Na and Mg. The predominance of K over the other minerals was already highlighted in other honeys—Moroccan and not—being a peculiar characteristic of such a bee product [33,40,41,58,63,64].

*G. alypum* honey from the Fquih Ben Salah province had the highest concentration of K (849.73 mg/Kg), while the lowest value of K was found in the sweet orange honey from Béni Mellal (102.80 mg/Kg). The highest Ca content was found in the PGI *Euphorbia* honey from Azilal (125.62 mg/Kg), and the lowest in sweet orange honey from Béni Mellal (81.70 mg/Kg). In jujube and *G. alypum* honeys, Na was the third mineral element with a concentration of 76.84 mg/Kg and 89.99 mg/Kg, respectively; while in sweet orange and PGI *Euphorbia* honeys, Mg was the third most abundant mineral, with a concentration of 65.46 mg/kg and 69.54 mg/Kg, respectively.

The mineral content of honeys under study agreed with the range of values reported for jujube and sweet orange honeys from the Béni Mellal-Khénifra region [40], as well as for the PGI Euphorbia honey [37,58].

For essential trace elements, the most significant contributions to the element profile came from Fe, Zn and Mn. Specifically, the highest concentration of Fe was found in *G. alypum* honey with a concentration of 16.51 mg/Kg, while the sweet orange honey had the lowest amount (6.89 mg/Kg). On the other hand, the PGI Euphorbia honey showed the most abundant concentrations of Mn and Zn (4.00 mg/Kg and 6.98 mg/Kg, respectively). Other essential trace elements (i.e., Cu, Se, Cr, Co, and Ni) were revealed at concentrations ≤1 mg/Kg.

Differently from major elements, no efforts have been devoted to the screening of trace elements in honey from the Béni Mellal-Kenifra region. However, Bettar et al. and Moujanni et al. recently revealed lower Fe and Mn contents for the PGI Euphorbia honey (respectively, 4.37–5.5 mg/Kg and <1 mg/Kg) [37,58].

Overall, it could be argued that the elemental profiles of different honeys are greatly affected by the floral source. Indeed, elements are primarily introduced from the soil into the nectar via the root system of the plant. Additionally, bees are in contact with the surrounding environment during foraging and further amounts of inorganic elements can be accidentally transferred from soil, water and soil to the hive. As a result, the elemental profile of honey is a bio-accumulative picture of the geographical context as well as of the activity near the apiary site [65,66].

### 3.2. Pesticide, PCB, and PAH Residues

No honey samples were shown to be free of pesticides. However, among the *n* = 108 pesticides investigated, no OCPs were revealed, and only *n* = 11 pesticides were found at levels higher than the respective LODs, mostly belonging to the organophosphate class (OPs) and its metabolites (Table 3). The jujube honey from the Khénifra province was among the samples with the highest number of quantifiable pesticides (*n* = 8), detected moreover at the highest levels. Such honey stood out for the highest level of carbaryl (1060.90 µg/Kg, *p* < 0.05), acephate (1251.19 µg/Kg, *p* < 0.05) and cyromazine (2060.99 µg/Kg, *p* < 0.05). Additionally, it was the only honey to show quantifiable residues of quinalphos (5.92 µg/Kg) and fenthion sulfoxide (16.53 µg/Kg). Intermediate and similar levels of contamination were found in the sweet orange honey produced in Béni Mellal and the PGI Euphorbia honey collected in Azilal, in which the most abundant residues were carbaryl (146.30 µg/Kg and 277.41 µg/Kg, *p* > 0.05) and cyromazine (223.72 µg/Kg and 113.60 µg/Kg, *p* > 0.05). Finally, the *G. alyphum* honey from the Fquih Ben Salah province had the lowest number of quantifiable pesticides (*n* = 5). In such honey, the most abundant residues were confirmed to be carbaryl, acephate and cyromazine. However, they were found at very low levels when compared with the other honey samples (*p* < 0.05). According to the Regulation (EC) No. 396/2005 and subsequent amendments [67], 75% of investigated samples (all samples from jujube, sweet orange, and the PGI Euphorbia honeys) widely exceeded the MRL of 0.05 mg/kg for carbaryl and cyromazine, as well as the MRL of 0.02 mg/Kg for acephate, while 50% of the samples (all samples from jujube and sweet orange honey) exceeded the MRLs of 0.01 mg/kg for dimethoate and diazinon. All samples of jujube honey greatly exceeded the MRLs of 0.01 mg/kg, 0.05 mg/kg, and 0.01 mg/kg respectively for alachlor, carbofuran and fenthion sulfoxide. As mentioned in the introduction section, very few efforts have been devoted to the assessment of the chemical safety of Moroccan honeys. A recent study conducted on the Euphorbia honey reported the identification and quantification of the 202 pesticides, including the ones detected in this study. However, contrasting results were obtained, since the detected residues were always within the set MRLs, thus indicating a good quality of the PGI product [42].

The pesticide fingerprint of different honeys clearly reflects the different agronomic practices of the different provinces of such Moroccan region and, more specifically, a more pronounced and prolonged use of OPs in the Khénifra province. The persistence of these pesticides on plants and soil can create shifts in the entire food chain. In fact, regardless of the type of honey, worker bees may transfer such contaminants from the pollen and nectar of plants to the hive, thus being inevitably incorporated into the different hive products [52].

Of the *n* = 18 PCBs under analysis, *n* = 2 compounds were found. Specifically, PCB118 was detected in all honey samples and quantified only in jujube honey from Khénifra province (0.71 µg/Kg, *p* < 0.05) and PCB180 was revealed in all types of honey and quantified in 75% of samples (0.42–0.73 µg/Kg, *p* < 0.05), apart from *G. alypum* honey from Fquih Ben Salah province (Table 3). Of the *n* = 13 PAHs investigated, *n* = 6 congeners were present in all samples, but quantified in 50% of them (i.e., jujube and sweet orange honeys). In particular, the jujube honey from Khénifra was the most contaminated product, with *n* = 5 PAHs detected at a level >LOQ. Between these, chrysene (2.10 µg/Kg), anthracene (1.54 µg/Kg) and fluorene (1.14 µg/Kg) were the most abundant toxicants. To follow, *n* = 4 PAHs were quantified in the sweet orange honey from Béni Mellal, with the most abundant compounds represented by benzo[a]anthracene (1.71 µg/Kg) and chrysene (1.62 µg/Kg) (Table 3). As previously mentioned, there are still no regulatory limits for PCBs and PAHs in honey and no toxicological consideration can be made in reference to the Reg. (EC) No. 1881/2006 [22], since it fixes the ML of just one PAH, namely the benzo[a]pyrene, and establishes a ML for the sum of PCBs, taking into account the share of fat in food.

However, the monitoring of PCBs and PAHs in Moroccan honey is very scarce. To the best knowledge of the authors, only Chakir et al. investigated diverse honey samples from different South, Center–South and East Moroccan regions and from many floral origins, including *C. sinensis* and *E. resiniphera* [68]. The study reported that a small share of samples was contaminated with PCBs, with concentration levels between 0.06 and 5.1 μg/kg. Furthermore, PAHs were present in all investigated samples with levels in the same range or slightly higher than those observed in this study (0.26–7.58 μg/kg). However, congeners such as dibenzo(a,h)anthracene and acenaphthylene were revealed at the highest levels.

A literature review pointed out that honey from the Mediterranean area produced during the last decade was poorly monitored with respect to pesticides, PCBs, and PAHs. In this respect, few recent works on Italian honey generally showed higher standards of chemical safety. In fact, organophosphorus pesticides were detected in the order of ng/g and not exceeding the relative MRLs; in addition, PAHs, such as acenaphthylene, fluorene, phenanthrene and pyrene, were found in the range >LOD-7.70 ng/g. PCBs were absent in all honeys investigated [52,67,69].

### 3.3. Plasticizers and BPs

Plasticizer and bisphenol residues revealed in the several honeys from the Béni Mellal-Khénifra region are shown in Table 4. Five PAEs (i.e., DEHP, DEP, DPrP, DiBP, and DBP) and five NPPs (i.e., DEA, DiBA, DBA, DEHA and DEHT) were determined at a concentration >LOQ in 100% samples. Among PAEs, DEP was the most abundant congener (0.94–3.17 mg/Kg *p* < 0.05) in the various honey, except for the *G. alypum* honey (0.94 mg/Kg, *p* < 0.05), followed by DBP (0.49–1.05 mg/Kg, *p* < 0.05) and DiBP (0.45–0.79 mg/Kg, *p* < 0.05) in all honey samples.

Among the NPPs, DBA was the most abundant compound (8.62–12.42 mg/Kg, *p* < 0.05) in the Moroccan honeys, except for the *G. alypum* honey (0.50 mg/Kg, *p* < 0.05), followed by DEA, with a concentration ranging between 1.30–5.65 mg/kg (*p* < 0.05). Furthermore, three BPs were also determined, namely BPA, BPB and BPAF. BPA was detected in 100% samples but quantified only in sweet orange and PGI Euphorbia honey samples (respectively, 7.74 µg/kg and 8.07 µg/kg, *p* > 0.05); BPAF was determined in all samples except the *G. Alypum* honey (1.48–158 µg/kg, *p* > 0.05); and BPB was the most abundant BP in all Moroccan honeys (4.16–8.75, *p* < 0.05). Among the determined plasticizers, Reg. (EU) No. 10/2011 has established SMLs from food contact material for DBP (0.3 mg/Kg), DEHP (1.5 mg/Kg), as well as DEP, DiBP and DEHT (60 mg/Kg) [23]. On this basis, all Moroccan honeys had an excessive amount of DBP, this PAE being at a concentration level 1.5–3 times higher than the relative SML. On the other hand, the regulatory SML for BPA migration from food contact material is 600 ng/g [24]. Table 4 shows that none of the honey samples contained BPA at concentrations higher than the SML.

Similarly to pesticides, PAHs and PCBs in the jujube honey from Khénifra and the sweet orange honey from Béni-Mellal demonstrated the highest levels of plasticizers and BPs, while the *G. alypum* honey from the Fquih Ben Salah province was generally the least contaminated product. However, in such honey, one PAE, i.e., DEHP, and one NPP, i.e., DEHT, were found at the highest levels (1.06 mg/kg and 1.14 mg/Kg, *p* < 0.05).

To the best of our knowledge, there is no literature regarding plasticizers and BPs in Moroccan honey. However, recent efforts can be observed in the determination of such chemicals in honey from the Mediterranean basin. In this respect, Lo Turco and coworkers [16] determined much lower levels of PAEs (e.g., DEP, 0.006 mg/Kg; DiBP, 0.042 mg/Kg; DBP, 0.039 mg/Kg; and DEHP, 0.191 mg/Kg) in Sicilian and Calabrian honeys than those determined in Moroccan honeys, and BPA was lower than analytical LOQ in all samples. More recently, Notardonato and colleagues [15] confirmed the higher purity of Italian honey, pointing out lower frequencies of PAE determination in honey samples, with only the DEHP found at concentrations (0.005–0.960) similar to those of the Moroccan honeys under study. Indeed, other PAEs, such as DEP (0.020–0.371 mg/Kg), DiBP (0.028–0.299 mg/Kg), DBP (0.019–0.550 mg/Kg) yielded much lower levels. However, Italian honeys revealed higher levels of BPA (18.8–996.8 µg/kg).

Since plasticizers and BPs could be easily released from the plastic components of honey production equipment (e.g., honey extractor and uncorkers), it may be assumed that the Moroccan honey could be contaminated by such compounds during production steps, as already observed not only in honey [15,16] but also in a variety of processed food [70,71]. Additionally, prolonged periods of storage with given conditions in terms of temperature, humidity, and light, may affect not only the peculiar physicochemical properties of honey, but also cause gradual polymer degradation, and, consequently, the migration of plastic additives from the plastic packaging into the honey [72]. However, due to the ubiquitous presence of plasticizers and BPs in the environment, the contamination of honey from the nectar source should always be considered [17].

### 3.4. Potentially Toxic Elements

The profile of potentially toxic trace elements of Moroccan honeys is reported in Table 5. Among investigated elements, Al was the most abundant metal and, differently from the trend of other contaminants, the highest and lowest levels of such metal were found respectively in the PGI Euphorbia honey from Azilal (5.56 mg/Kg, *p* < 0.05) and in the jujube honey produced in Khénifra (1.69 mg/Kg, *p* < 0.05).

Aluminum may be related to the soil acidification caused by elevated industrial emissions and poor mining practices occurring in such Moroccan regions, and it may be increasingly bioavailable to organisms through plant roots. Inevitably, the metal can then spread up the food chain through pollen, and nectar collection [73].

To follow, equal amounts of Pb and As (0.06–0.16 mg/Kg, *p* > 0.05) were determined in all samples, while Cd was lower than LOQ (0.003 µg/Kg) in any case. Regulation (EU) No. 2015/1005, amending Regulation (EC) No. 1881/2006 for the ML of Pb in certain foods [74], has introduced a ML of 0.1 mg/Kg in honey. Accordingly, 3 out of the 4 types of Moroccan honey, namely the sweet orange, the PGI Euphorbia and the *G. alypum* honeys, were characterized by a Pb content higher than the regulatory limit.

Anthropogenic activities, including industrial applications and agricultural chemicals, are responsible for the heavy metal contamination of surroundings. As with other contaminants, bees and bee products including honey are exposed to these contaminants via polluted pollen, water and air. The role of honeybees as “filters” of heavy metals and their protective function against honey contamination have been commonly accepted [75,76]. In contrast, honey is still considered as a typical indicator of heavy metal pollution related not only to anthropogenic activities (e.g., agriculture, industry etc.) but also to the entire production process, as poor beekeeping practices may be also source of heavy metal residues in honey [77,78].

In the last decade, the issue of potentially toxic trace elements in honey from Mediterranean countries has been approached and, as expected, great data variability was pointed out. For example, carob honey from different Moroccan areas and the PGI Euphorbia honey did not display heavy metals such as Cd and Pb, probably because of the high instrumental LOQ values [41,42]. Honey from several Libyan locations had very high levels of Pb (2.42–10.98 mg/Kg), Cd (0.125–0.150 mg/Kg), but low As contents (0.006–0.018 mg/Kg) [79]. Despite honeys from different Italian regions generally showing Cd and Al at higher levels than Moroccan honeys (respectively, 0.003–0.02 mg/Kg and 0.52–26 mg/Kg) [80,81,82,83,84], Pb was always under the regulatory limit (0.05–0.06 mg/Kg) [79,80] and As at lower levels than those detected in this study (0.01 mg/Kg) [81,84]. On the other hand, different varieties of Spain honeys were marked by lower levels of As (0.004–0.010 mg/Kg) and Pb (0.011–0.041 mg/Kg) [85].

### 3.5. PCA Analysis

In the present study, PCA provided information on the most significant variables describing the whole data set, enabling data reduction at the same time with a minimum loss of original information.

Four principal components (PCs) with eigenvalues exceeding one (325.627, 18.195, 7.575 and 1.257) were extracted according to the Kaiser Criterion, and they explained up to 97.51% of total variance (i.e., 47.457%, 33.695%, 14.029% and 2.327%, respectively). Figure 2 illustrates the bidimensional score and loading plots. Defined by the first two PCs accounting for more than 81% of the variability of the system, the score plot (Figure 2left) showed four distinguished clusters of honey samples. Such clusters correspond to the four types of Moroccan honey investigated, which differed from each other not only in their botanical origin (i.e., jujube, sweet orange, Euphorbia and *G. alypum* honeys) but also for the production area (i.e., provinces of Khénifra, Béni Mellal, Azilal and Fquih Ben Salah). Due to such a sample arrangement in the study, it is somewhat troublesome to define whether the clustering of honeys occurred according the botanical or geographic origin. However, based on the array of variables investigated, the obtained data and the provided considerations, it may be argued that both factors noticeably contributed to such sample differentiation. Indeed, the loading plot (Figure 2right) shows that, in accordance with the results of the Kruskal–Wallis test, almost all investigated parameters (i.e., physicochemical indicators, minerals, organic and inorganic contaminants), each related to the floral and/or geographical origin, contributed significantly to the differentiation of honey samples, except for pH, DEHT, DEHA, BPAF, Pb and As.

By overlapping the loading and score plots, it becomes clear that variables such as pesticides, PAHs, plasticizers and BPs weighed more on the jujube honey from Khénifra and the sweet orange honey from Béni Mellal which, in fact, were the honey samples most affected by organic contamination.

The PGI Euphorbia honey was in general less contaminated than the above honeys, but it was still characterized by the highest content of pesticides, such as diazinon and metalaxyl-M, and potentially toxic metals, such as Al.

On the other hand, the *G. alypum* honey was the least contaminated Moroccan product and, moreover, it was marked by the most convenient physicochemical traits (i.e., TSS, acidity and conductivity), as well as precious contents of minerals and essential trace elements.

### 3.6. Dietary Exposure to Contaminants

The quality of Moroccan honeys and the potential health risk to consumers were assessed by calculating the EDI and the non-carcinogenic risk (HQ) of organic and inorganic contaminants (Table 6).

EDIs and HQs were calculated by considering the amount of honey consumed daily in the diet by an adult consumer (70 Kg) from Europe (1.8 g/day) and North Africa (0.3 g/day), according to FAO [86], as well as guideline values recommended by international organizations (Appendix A).

As shown in Table 6, the EDIs calculated were well below the intake levels of relative pollutants recommended by international regulatory bodies, thus indicating that Moroccan honey can be safely consumed through the provided dietary amounts. For the non-carcinogenic risk assessment, HQ did not exceed the threshold value of 1 for each contaminant potentially ingested by adults through honey in both European and North-African diets, thus indicating that non-carcinogenic health effects derived from the consumption of these Moroccan honeys were not significant.

## 4. Conclusions

For the first time, a comprehensive characterization of the physicochemical traits and contaminants of four monofloral honeys from different provinces of the Moroccan region Béni Mellal-Khénifra was carried out, thus corroborating the scarce literature on Moroccan honey. According to the physicochemical parameters, all honeys under analysis were in line with those EU standards established for assuring the authenticity of such bee products. However, a critical contamination pattern was outlined, with several toxicants often exceeding the EU regulatory limits available for honey. Specifically, the jujube honey from Khénifra and the sweet orange honey from the Béni Mellal province were the most contaminated products, as opposed to the *G. alypum* honey from the Fquih Ben Salah province, which was shown to be the least contaminated one. In this arrangement, the PGI Euphorbia honey from the Azilal area had an intermediate contamination degree. However, the dietary exposure assessment highlighted that small amounts of all honeys can be safely introduced both in European and North African diets on a daily basis.

Hopefully, findings from this study should not only encourage the enforcement and harmonization of the international regulatory framework on the chemical safety of honey, which is still suffering from evident flaws and shortcomings, but also serve as a “wake-up call” for Moroccan governmental bodies to strengthen monitoring activity in beekeeping and to find suitable solutions for accomplishing more sustainable agricultural practices and modernize the beekeeping sector.

## Figures and Tables

**Figure 1 foods-12-00969-f001:**
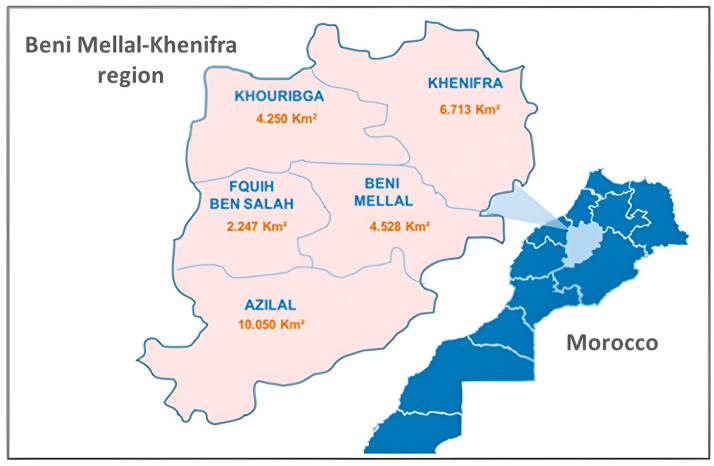
Geographical map of the Moroccan region Béni Mellal-Khénifra and its provinces.

**Figure 2 foods-12-00969-f002:**
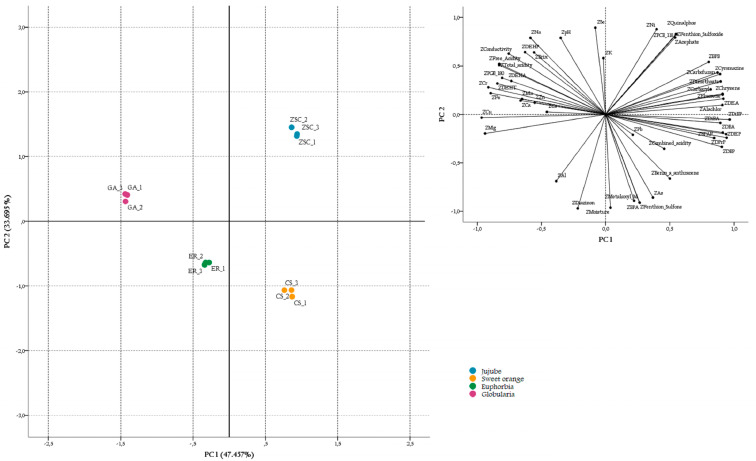
PCA score (**left**) and loading (**right**) plots of PC1 and PC2 showing the differentiation of honey samples in the component space based on the array of variables investigated.

**Table 1 foods-12-00969-t001:** Physicochemical traits of monofloral honeys from the Béni Mellal-Khénifra region. Values are expressed as mean ± standard deviation of *n* = 3 samples. Statistics from Kruskal–Wallis test are also reported.

Parameter	Jujube Honey (Khénifra)	Sweet Orange Honey (Béni Mellal)	Euphorbia Honey (PGI, Azilal)	*G. alypum* Honey (Fquih Ben Salah)	F Statistic	*p*-Value
**Moisture (%)**	14.93 ^a^ ± 0.15	16.57 ^b^ ± 0.21	15.47 ^ab^ ± 0.25	16.40 ^b^ ± 0.26	9.667	**0.022**
**TSS (°Brix)**	85.00 ^ab^ ± 1.00	82.67 ^a^ ± 0.58	84.27 ^ab^ ± 0.64	85.83 ^b^ ± 1.26	8.048	**0.045**
**Free acidity (meq/kg)**	25.46 ^a^ ± 0.15	15.41 ^b^ ± 0.07	27.49 ^a^ ± 0.34	39.28 ^c^ ± 0.85	10.395	**0.016**
**Combined acidity (meq/kg)**	0.99 ^a^ ± 0.01	0.98 ^a^ ± 0.01	1.50 ^b^ ± 0.02	0.60 ^c^ ± 0.10	9.721	**0.021**
**Total acidity (meq/kg)**	26.45 ^a^ ± 0.16	16.39 ^b^ ± 0.08	28.98 ^a^ ± 0.33	39.88 ^c^ ± 0.83	10.385	**0.016**
**pH**	4.24 ± 0.08	4.24 ± 0.06	4.10 ± 0.10	3.98 ± 0.06	7.307	0.063

a–c indicate homogeneous sample groups at α = 0.05 and honeys which do not differ from each other are designated by the same letter. Bold *p*-values showed significantly different results at *p* < 0.05 between different honeys.

**Table 2 foods-12-00969-t002:** Conductivity, ash and profile of minerals and essential trace elements of monofloral honeys from the Béni Mellal-Khénifra region. Values are expressed as mean ± standard deviation of *n* = 3 samples. Statistics from the Kruskal–Wallis test are also reported.

Parameter	Jujube Honey (Khénifra)	Sweet Orange Honey (Béni Mellal)	Euphorbia Honey (PGI, Azilal)	*G. alypum* Honey (Fquih Ben Salah)	F Statistic	*p*-Value
**Conductivity (µS/cm)**	381.33 ^a^ ± 18.23	157.00 ^b^ ± 1.00	362.67 ^a^ ± 8.62	633.67 ^c^ ± 8.02	9.974	**0.019**
**Ash (g/Kg)**	1.17 ^a^ ± 0.18	0.34 ^b^ ± 0.05	0.97 ^a^ ± 0.08	1.51 ^c^ ± 0.10	9.112	**0.025**
**K (mg/Kg)**	753.25 ^a^ ± 72.86	102.80 ^b^ ± 2.71	695.87 ^a^ ± 38.61	849.73 ^c^ ± 81.83	9.667	**0.022**
**Ca (mg/Kg)**	98.18 ^a^ ± 0.89	81.70 ^a^ ± 0.80	125.62 ^b^ ± 1.00	110.75 ^b^ ± 1.68	10.385	**0.016**
**Na (mg/Kg)**	76.84 ^a^ ± 0.32	35.21 ^b^ ± 0.67	60.41 ^a,c^ ± 1.06	89.99 ^a^ ± 2.14	10.385	**0.016**
**Mg (mg/Kg)**	57.87 ^a^ ± 2.42	65.46 ^ab^ ± 0.94	69.54 ^ab^ ± 0.91	85.24 ^b^ ± 1.67	10.385	**0.016**
**Mn (mg/Kg)**	1.35 ^a^ ± 0.11	0.57 ^a^ ± 0.03	4.00 ^b^ ± 0.11	2.24 ^ab^ ± 0.12	10.385	**0.016**
**Fe (mg/Kg)**	9.50 ^a^ ± 0.63	6.89 ^a^ ± 0.10	14.34 ^b^ ± 0.29	16.51 ^b^ ± 0.24	10.385	**0.016**
**Zn (mg/Kg)**	2.81 ^ab^ ± 0.06	1.41 ^a^ ± 0.03	6.98 ^c^ ± 0.51	3.60 ^b^ ± 0.18	10.385	**0.016**
**Cu (mg/Kg)**	0.27 ^a^ ± 0.09	0.27 ^a^ ± 0.06	0.86 ^b^ ± 0.06	1.59 ^b^ ± 0.18	9.539	**0.025**
**Se mg/Kg)**	0.25 ^a^ ± 0.03	0.09 ^b^ ± 0.01	0.16 ^c^ ± 0.01	0.17 ^c^ ± 0.01	9.462	**0.024**
**Cr (mg/Kg)**	0.09 ^a^ ± 0.01	0.06 ^a^ ± 0.01	0.11 ^a^ ± 0.01	0.94 ^b^ ± 0.05	10.385	**0.016**
**Co (mg/Kg)**	0.04 ^ab^ ± 0.01	0.01 ^a^ ± 0.01	0.24 ^b^ ± 0.11	0.05 ^ab^ ± 0.02	9.585	**0.022**
**Ni (mg/Kg)**	0.54 ^a^ ± 0.11	0.17 ^b^ ± 0.04	0.17 ^b^ ± 0.01	0.21 ^b^ ± 0.03	8.231	**0.041**

a–c indicate homogeneous sample groups at α = 0.05 and honeys which do not differ from each other are designated by the same letter. Bold *p*-values showed significantly different results at *p* < 0.05 between different honeys.

**Table 3 foods-12-00969-t003:** Residues of pesticides, PCBs, and PAHs detected in several honeys from the Béni Mellal-Khénifra region. Data are expressed as mean ± standard deviation of *n* = 3 samples analyzed per honey. Statistics from the Kruskal–Wallis test are also reported.

Analyte (µg/kg)	Jujube Honey (Khénifra)	Sweet Orange Honey (Béni Mellal)	Euphorbia Honey (PGI, Azilal)	*G. alypum* Honey (Fquih Ben Salah)	F Statistic	*p*-Value
**Carbaryl**	1060.90 ^a^ ± 71.34	146.30 ^b^ ± 7.24	277.41 ^b^ ± 23.24	16.62 ^c^ ± 0.70	10.385	**0.016**
**Dimethoate**	72.01 ^a^ ± 4.92	14.31 ^b^ ± 1.28	<LOQ	<LOQ	10.649	**0.014**
**Carbofuran**	77.30 ^a^ ± 1.90	5.30 ^b^ ± 1.00	<LOQ	<LOQ	10.649	**0.014**
**Diazinon**	<LOQ	25.50 ^a^ ± 0.81	27.31 ^a^ ± 1.56	2.41 ^b^ ± 0.35	10.116	**0.018**
**Alachlor**	22.66 ^a^ ± 1.21	9.05 ^b^ ± 0.24	8.75 ^b^ ± 0.44	1.37 ^c^ ± 0.17	9.667	**0.022**
**Metalaxyl-M**	<LOQ	11.43 ^b^ ± 1.19	28.35 ^c^ ± 1.10	<LOQ	10.649	**0.014**
**Quinalphos**	5.92 ^a^ ± 0.30	<LOQ	<LOQ	<LOQ	10.735	**0.013**
**Fenthion Sulfoxide**	16.53 ^a^ ± 1.05	<LOQ	<LOQ	<LOQ	10.735	**0.013**
**Fenthion Sulfone**	<LOQ	5.26 ^a^ ± 0.74	6.11 ^a^ ± 3.25	<LOQ	9.598	**0.022**
**Acephate**	1251.19 ^a^ ± 147.67	11.46 ^b^ ± 0.69	25.49 ^b^ ± 1.61	11.45 ^b^ ± 1.25	9.359	**0.025**
**Cyromazine**	2060.99 ^a^ ± 75.05	223.72 ^b^ ± 12.71	113.60 ^b^ ± 3.86	21.72 ^c^ ± 1.48	10.385	**0.016**
**PCB118**	0.71 ^a^ ± 0.01	<LOQ	<LOQ	<LOQ	10.800	**0.013**
**PCB180**	0.72 ^a^ ± 0.07	0.43 ^b^ ± 0.03	0.42 ^b^ ± 0.04	<LOQ	9.565	**0.023**
**Acenaphthylene**	0.66 ^a^ ± 0.05	< LOQ	<LOQ	<LOQ	10.735	**0.013**
**Fluorene**	1.14 ^a^ ± 0.10	0.86 ^a^ ± 0.05	<LOQ	<LOQ	10.649	**0.014**
**Phenanthrene**	0.65 ^a^ ± 0.04	<LOQ	<LOQ	<LOQ	10.735	**0.013**
**Anthracene**	1.54 ^a^ ± 0.11	<LOQ	<LOQ	<LOQ	10.735	**0.013**
**Benzo[a]anthracene**	<LOQ	1.71 ^a^ ± 0.09	<LOQ	<LOQ	10.735	**0.013**
**Chrysene**	2.10 ^a^ ± 0.06	1.62 ^a^ ± 0.06	<LOQ	< LOQ	10.649	**0.014**

a–c indicate homogeneous sample groups at α = 0.05 and honeys which do not differ from each other are designated by the same letter. Bold *p*-values showed significantly different results at *p* < 0.05 between different honeys.

**Table 4 foods-12-00969-t004:** Plasticizers (PAEs and NPPs) and BPs detected in several honeys from the Béni Mellal-Khénifra region. Data are expressed as mean ± standard deviation of *n* = 3 samples analyzed per honey. Statistics from the Kruskal–Wallis test are also reported.

Analyte	Jujube Honey (Khénifra)	Sweet orange Honey (Béni Mellal)	Euphorbia Honey (PGI, Azilal)	*G. alypum* Honey (Fquih Ben Salah)	F Statistic	*p*-Value
Plasticizers (mg/Kg)						
**DEHP**	0.59 ^ab^ ± 0.03	0.36 ^a^ ± 0.02	0.35 ^a^ ± 0.03	1.06 ^b^ ± 0.03	9.359	**0.025**
**DEP**	2.95 ^a^ ± 0.10	3.1 ^a^ ± 0.05	2.42 ^a^ ± 0.28	0.94 ^b^ ± 0.06	10.385	**0.016**
**DPrP**	0.61 ^a^ ± 0.03	0.65 ^a^ ± 0.02	0.57 ^a^ ± 0.01	0.42 ^b^ ± 0.01	9.974	**0.019**
**DiBP**	0.77 ^a^ ± 0.04	0.79 ^a^ ± 0.03	0.60 ^ab^ ± 0.07	0.45 ^b^ ± 0.02	9.359	**0.025**
**DBP**	0.89 ^a^ ± 0.06	1.05 ^a^ ± 0.05	0.84 ^a^ ± 0.04	0.49 ^b^ ± 0.04	9.667	**0.022**
**DEA**	5.65 ^a^ ± 0.48	5.30 ^a^ ± 0.07	1.35 ^b^ ± 0.09	1.30 ^b^ ± 0.16	8.436	**0.038**
**DiBA**	0.85 ^a^ ± 0.05	0.82 ^a^ ± 0.03	0.76 ^a^ ± 0.01	0.53 ^b^ ± 0.08	9.462	**0.024**
**DBA**	12.42 ^a^ ± 0.31	12.23 ^a^ ± 0.40	8.62 ^b^ ± 0.65	0.50 ^c^ ± 0.04	9.667	**0.022**
**DEHA**	0.39 ± 0.06	0.36 ± 0.08	0.36 ± 0.05	0.63 ± 0.05	9.368	0.095
**DEHT**	0.59 ± 0.03	0.52 ± 0.07	0.58 ± 0.09	1.14 ± 0.10	6.897	0.075
Bisphenols (µg/kg)						
**BPA**	<LOQ	7.74 ^b^ ± 0.69	8.07 ^b^ ± 0.08	<LOQ	9.598	**0.022**
**BPB**	8.75 ^b^ ± 1.15	5.72 ^ab^ ± 0.61	4.62 ^a^ ± 0.31	4.16 ^a^ ± 0.10	10.385	**0.016**
**BPAF**	1.48 ± 0.17	1.55 ± 0.24	1.58 ± 0.17	<LOQ	6.668	0.083

a–c indicate homogeneous sample groups at α = 0.05 and honeys which do not differ from each other are designated by same letter. Bold *p*-values showed significantly different results at *p* < 0.05 between different honeys.

**Table 5 foods-12-00969-t005:** Potentially toxic elements of several honeys from the Béni Mellal-Khénifra region. Data are expressed as mean ± standard deviation of *n* = 3 samples analyzed per honey. Statistics from the Kruskal–Wallis’s test are also reported.

Analyte (mg/Kg)	Jujube Honey (Khénifra)	Sweet Orange Honey (Béni Mellal)	Euphorbia Honey (PGI, Azilal)	*G. alypum* Honey (Fquih Ben Salah)	F Statistic	*p*-Value
**Al**	1.69 ^a^ ± 0.15	2.73 ^ab^ ± 0.19	5.56 ^b^ ± 0.12	2.67 ^ab^ ± 0.12	9.359	**0.025**
**Pb**	0.06 ± 0.01	0.16 ± 0.06	0.15 ± 0.04	0.12 ± 0.03	3.873	0.276
**Cd**	<LOQ	<LOQ	<LOQ	<LOQ	<LOQ	0.108
**As**	0.06 ± 0.01	0.11 ± 0.04	0.09 ± 0.03	0.06 ± 0.02	9.430	**0.249**

a–c indicate homogeneous sample groups at α = 0.05 and honeys which do not differ from each other are designated by the same letter. Bold *p*-values showed significantly different results at *p* < 0.05 between different honeys.

**Table 6 foods-12-00969-t006:** EDIs (mg/Kg_bw_/day) and HQs calculated for investigated Moroccan honeys daily consumed by adult consumers both from Europe and North Africa.

Contaminant	Jujube Honey (Khénifra)				Sweet Orange Honey (Béni Mellal)				Euphorbia Honey (PGI, Azilal)				*G. alypum* Honey (Fquih Ben Salah)			
	Europe		North Africa		Europe		North Africa		Europe		North Africa		Europe		North Africa	
	EDI	HQ	EDI	HQ	EDI	HQ	EDI	HQ	EDI	HQ	EDI	HQ	EDI	HQ	EDI	HQ
	Plasticizers and BPs															
**DEHP**	1.52 × 10^5^	<1	2.53 × 10^6^	<1	9.25 × 10^6^	<1	1.54 × 10^6^	<1	9.00 × 10^6^	<1	1.50 × 10^6^	<1	2.73 × 10^5^	<1	4.54 × 10^6^	<1
**DEP**	7.59 × 10^5^	<1	1.26 × 10^5^	<1	7.97 × 10^5^	<1	1.32 × 10^5^	<1	6.22 × 10^5^	<1	1.04 × 10^5^	<1	2.42 × 10^5^	<1	4.03 × 10^6^	<1
**DBP**	2.29 × 10^5^	<1	3.81 × 10^6^	<1	0.27 × 10^5^	<1	0.45 × 10^5^	<1	0.21 × 10^4^	<1	3.60 × 10^6^	<1	1.26 × 10^5^	<1	2.10 × 10^6^	<1
**DEHA**	1.00 × 10^5^	<1	1.67 × 10^6^	<1	9.25 × 10^6^	<1	1.54 × 10^6^	<1	9.25 × 10^6^	<1	1.54 × 10^6^	<1	1.62 × 10^5^	<1	2.70 × 10^6^	<1
**BPA ***	-	-	-	-	0.20 × 10^3^	<1	3.31 × 10^5^	<1	0.21 × 10^3^	<1	3.46 × 10^5^	<1	-	-	4.54 × 10^6^	-
	Potentially toxic elements															
**Al ***	3.04 × 10^4^	<1	5.07 × 10^5^	<1	4.91 × 10^4^	<1	8.19 × 10^5^	<1	1.00 × 10^3^	<1	1.67 × 10^4^	<1	4.81 × 10^4^	<1	8.01 × 10^5^	<1
**Pb**	1.54 × 10^6^	<1	2.57 × 10^7^	<1	4.11 × 10^6^	<1	6.86 × 10^7^	<1	3.86 × 10^6^	<1	6.43 × 10^7^	<1	3.09 × 10^6^	<1	5.14 × 10^7^	<1
**As**	1.54 × 10^3^	<1	2.57 × 10^4^	<1	2.83 × 10^3^	<1	4.71 × 10^4^	<1	2.31 × 10^3^	<1	3.86 × 10^4^	<1	1.54 × 10^3^	<1	2.57 × 10^4^	<1

* EDI Expressed as µg/Kgbw/day.

## Data Availability

Data is contained within the article or Appendix A.

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
