# Peer review of "Monitoring Moroccan Honeys: Physicochemical Properties and Contamination Pattern"

_foods, 2023, doi:10.3390/foods12050969_

Round 1

Reviewer 1 Report

Dear Editor,

It is an interesting study aimed to evaluate the physicochemical traits and an array of organic and inorganic contaminants in monofloral honeys in Marocco.

I have some suggestions to improve quality of the paper.

Line 19. PAHs – Please, give a full name of this chemical and put abbreviation in brackets - Polycyclic aromatic hydrocarbons (PAHs).

Line 20. DBP – see above statement.

The authors do not mention data regarding the most widely used in recent years, a new generation of insecticides – neonicotinoids.

The authors mention many European laws and regulations that control the presence of xenobiotics in honey, but do not mention anything about the agricultural practices that exist in Béni Mellal-Khénifra region - what pesticides, insecticides etc. are used in the treatment of agricultural crops or seeds.

Line 133-136. Please, put this in the Introduction section.

Line 138. Please, clarify whether these samples are pooled or from single hive!

Line 277-278. What does mean LOQ and LOD value? Please, give a full name of these abbreviations?

Line 325-326. How the authors explain the lowest pH in G. alypum honey and vice versa highest values in sweet orange and jujube honey?

Line 343. Please, give more information about higher electrical conductivity honeydew honeys.

Do the authors have any explanation for the different mineral composition in the different honey samples? And is there association between mineral content and botanical origin of honey?

Line 392-419. What are the reasons for the observed differences regarding residues of pesticides in different botanical honey samples?

Line 436. Chakir and colleagues – replace with Chakir et al. [70].

Line 486, 490. Lo Turco and coworkers [16] and Notardonato and colleagues – Please, see above recommendation.

Line 514. Al was the most abundant metal – Please, do not begin the sentence with this metal, it is confusing for readers.

Line 514. What, according to the authors, accounts for the high aluminum content in the honey samples? Is there any objective reason for this?

The references do not follow the requirements.

Author Response

Dear Editor,

It is an interesting study aimed to evaluate the physicochemical traits and an array of organic and inorganic contaminants in monofloral honeys in Marocco.

I have some suggestions to improve quality of the paper.

R:/ thank you for the positive comment. We accepted the reviewer’s suggestions and we hope of improving the overall quality of the manuscript.

Line 19. PAHs – Please, give a full name of this chemical and put abbreviation in brackets - Polycyclic aromatic hydrocarbons (PAHs).

Line 20. DBP – see above statement.

R.:/ Accepted

The authors do not mention data regarding the most widely used in recent years, a new generation of insecticides – neonicotinoids.

R.:/ That’s true. Unfortunately, at the time of the study we lack commercial standards of neonicotinoids useful for studying these pesticides in honey.

The authors mention many European laws and regulations that control the presence of xenobiotics in honey, but do not mention anything about the agricultural practices that exist in Béni Mellal-Khénifra region - what pesticides, insecticides etc. are used in the treatment of agricultural crops or seeds.

Line 133-136. Please, put this in the Introduction section.

R.:/Thank you for the suggestion. Due to the specificity and the relevance of the content we would prefer to keep the description of the region and of its “agricultural status” in the paragraph 2.1

Line 138. Please, clarify whether these samples are pooled or from single hive!

R.:/ each honey was produced from different hives. It has been specified.

Line 277-278. What does mean LOQ and LOD value? Please, give a full name of these abbreviations?

R.:/ corrected

Line 325-326. How the authors explain the lowest pH in G. alypum honey and vice versa highest values in sweet orange and jujube honey?

R.:/ In line 314-316 was stated: “Honey acidity and pH are correlated with each other, being dependent on the level of organic acids and enzymatic activity in honey. As a result, their variation in honey samples could be attributed to the floral origin rather than the environmental context [rif]”. Based on this assumption different pH value may be related to the different floral origin of samples. It has been added in line 326-327

Line 343. Please, give more information about higher electrical conductivity honeydew honeys.

R.:/More information has been added in lines 344-347.

Do the authors have any explanation for the different mineral composition in the different honey samples? And is there association between mineral content and botanical origin of honey?

R.:/A potential explanation to the elemental data has been added in lines 390-396.

Line 392-419. What are the reasons for the observed differences regarding residues of pesticides in different botanical honey samples?

R.:/A potential explanation to obtained results has been added in lines 433-437

Line 436. Chakir and colleagues – replace with Chakir et al. [70].

R:/ Done

Line 486, 490. Lo Turco and coworkers [16] and Notardonato and colleagues – Please, see above recommendation.

R.:/ to avoid repetition the authors would like to maintain the used prepositions

Line 514. Al was the most abundant metal – Please, do not begin the sentence with this metal, it is confusing for readers.

R.:/ Corrected

Line 514. What, according to the authors, accounts for the high aluminum content in the honey samples? Is there any objective reason for this?

R.:/A potential explanation to obtained results has been added in lines 537-540 e 547-554

Reviewer 2 Report

This is an interesting and comprehensive work, characterizing four types of Moroccan honeys with focus on their physicochemical traits and contaminants, including potentially toxic components. 

The authors provided sufficient background in the Introduction section. The applied methods are appropriate and scientifically sound. The results are presented in a clear way, supported by useful summarizing tables and a PCA plot. However, the quality of Figure 2 should be improved, because it is hardly readable in its current form (at least font size of legend should be larger).
A novel approach was provided by assessing the potential health risk of consuming various honeys, taking into account typical dietary intakes.

Below please find some detailed comments and suggestions related mostly to language issues:

L 115: ‘consumption’ should be used instead of “assumption”

L 145: What does a “fresh” place mean? Maybe it is enough to state that the storage place was dark and at room temperature.

L 311: ‘inversely related’ instead of “inversed related”

L 322, L 346-347: ‘significant differences’ instead of “significantly different”

L 535: ‘in contrast’ seems to be appropriate here instead of “despite”

L 579: delete “as well as” from the end of the sentence!

L 593 and L 599: ‘consumed’ instead of “assumed”

L 604: ‘four’ instead of “several” would be more realistic

L 611: ‘as opposed to’ instead of “as opposed as”

On the whole, I liked this manuscript and also believe that it can be of great interest to readers of Foods.

Author Response

This is an interesting and comprehensive work, characterizing four types of Moroccan honeys with focus on their physicochemical traits and contaminants, including potentially toxic components. 

The authors provided sufficient background in the Introduction section. The applied methods are appropriate and scientifically sound. The results are presented in a clear way, supported by useful summarizing tables and a PCA plot. However, the quality of Figure 2 should be improved, because it is hardly readable in its current form (at least font size of legend should be larger).
A novel approach was provided by assessing the potential health risk of consuming various honeys, taking into account typical dietary intakes.

R.:/Thank you for your positive comments. The authors have accepted all your suggestions in the hope of improving the overall quality of the manuscript. Concerning the figure with PCA. A high-resolution figure has been uploaded into the submission system of Foods.

Below please find some detailed comments and suggestions related mostly to language issues:

L 115: ‘consumption’ should be used instead of “assumption”

R.:/ Corrected

L 145: What does a “fresh” place mean? Maybe it is enough to state that the storage place was dark and at room temperature.

R.:/ Corrected

L 311: ‘inversely related’ instead of “inversed related”

R.:/ Corrected

L 322, L 346-347: ‘significant differences’ instead of “significantly different”

R.:/ Corrected

L 535: ‘in contrast’ seems to be appropriate here instead of “despite”

R.:/ Corrected

L 579: delete “as well as” from the end of the sentence!

R.:/ Corrected

L 593 and L 599: ‘consumed’ instead of “assumed”

R.:/ Corrected

L 604: ‘four’ instead of “several” would be more realistic

R.:/ Corrected

L 611: ‘as opposed to’ instead of “as opposed as”

R.:/as opposed to’

On the whole, I liked this manuscript and also believe that it can be of great interest to readers of Foods.

Reviewer 3 Report

It is an up-to-date paper about the characterization of honey from Morocco with emphasis on pesticide contamination in the context of the need to reduce pollution but also to ensure traceability. The work is complex with important results both for the field of research and for the development of policies in the field. Specifically, I think that the abstract could be improved, to be clearer and more concrete and uniform, for example in the abstract some plants have the scientific name while others do not, you should put the scientific name for all species. I also think that in methods 2.2 Honey samples and Chemicals the expression should be clearer.

Author Response

It is an up-to-date paper about the characterization of honey from Morocco with emphasis on pesticide contamination in the context of the need to reduce pollution but also to ensure traceability. The work is complex with important results both for the field of research and for the development of policies in the field. Specifically, I think that the abstract could be improved, to be clearer and more concrete and uniform, for example in the abstract some plants have the scientific name while others do not, you should put the scientific name for all species. I also think that in methods 2.2 Honey samples and Chemicals the expression should be clearer.

R.:/ Thank you for the positive comment.

In the abstract, the names of botanical species have been added. Furthermore section 2.2 has been improved by adding more precise information on honey samples and their storage. The section “Chemicals” has been changed to “Chemicals and reagents”. Plus, the enumeration of subsections present in Materials and methods has been reviewed.